

# Sympathy for the devil: a conservation strategy for devil and manta rays

Julia M. Lawson[1], Sonja V. Fordham[2], Mary P. O'Malley[3,4], Lindsay N.K. Davidson[1], Rachel H.L. Walls[1], Michelle R. Heupel[5], Guy Stevens[4,6], Daniel Fernando[4,7,8], Ania Budziak[9], Colin A. Simpfendorfer[10], Isabel Ender[4], Malcolm P. Francis[11], Giuseppe Notarbartolo di Sciara[12] and Nicholas K. Dulvy[1]

[1] Earth to Ocean Research Group, Department of Biological Sciences, Simon Fraser University, Burnaby, British Columbia, Canada
[2] Shark Advocates International, The Ocean Foundation, Washington, D.C., United States of America
[3] WildAid, San Francisco, CA, United States of America
[4] Manta Trust, Dorchester, Dorset, United Kingdom
[5] Australian Institute of Marine Science, Townsville, Queensland, Australia
[6] Environment Department, University of York, York, United Kingdom
[7] Department of Biology and Environmental Science, Linnaeus University, Kalmar, Sweden
[8] Blue Resources, Colombo, Sri Lanka
[9] Project AWARE Foundation, Rancho Santa Margarita, CA, United States of America
[10] Centre for Sustainable Tropical Fisheries and Aquaculture & College of Marine and Environmental Sciences, James Cook University, Townsville, Queensland, Australia
[11] National Institute of Water and Atmospheric Research, Wellington, New Zealand
[12] Tethys Research Institute, Milan, Italy

Corresponding author
Nicholas K. Dulvy, dulvy@sfu.ca

## ABSTRACT

**Background**. International trade for luxury products, medicines, and tonics poses a threat to both terrestrial and marine wildlife. The demand for and consumption of gill plates (known as *Peng Yu Sai*, "Fish Gill of Mobulid Ray") from devil and manta rays (subfamily Mobulinae, collectively referred to as mobulids) poses a significant threat to these marine fishes because of their extremely low productivity. The demand for these gill plates has driven an international trade supplied by largely unmonitored and unregulated catches from target and incidental fisheries around the world. Scientific research, conservation campaigns, and legal protections for devil rays have lagged behind those for manta rays despite similar threats across all mobulids.

**Methods**. To investigate the difference in attention given to devil rays and manta rays, we examined trends in the scientific literature and updated species distribution maps for all mobulids. Using available information on target and incidental fisheries, and gathering information on fishing and trade regulations (at international, national, and territorial levels), we examined how threats and protective measures overlap with species distribution. We then used a species conservation planning approach to develop the Global Devil and Manta Ray Conservation Strategy, specifying a vision, goals, objectives, and actions to advance the knowledge and protection of both devil and manta rays.

**Results and Discussion**. Our literature review revealed that there had been nearly 2.5-times more "manta"-titled publications, than "mobula" or "devil ray"-titled publications over the past 4.5 years (January 2012–June 2016). The majority of these recent publications were reports on occurrence of mobulid species. These publications

contributed to updated Area of Occupancy and Extent of Occurrence maps which showed expanded distributions for most mobulid species and overlap between the two genera. While several international protections have recently expanded to include all mobulids, there remains a greater number of national, state, and territory-level protections for manta rays compared to devil rays. We hypothesize that there are fewer scientific publications and regulatory protections for devil rays due primarily to perceptions of charisma that favour manta rays. We suggest that the well-established species conservation framework used here offers an objective solution to close this gap. To advance the goals of the conservation strategy we highlight opportunities for parity in protection and suggest solutions to help reduce target and bycatch fisheries.

## INTRODUCTION

International trade poses an increasing threat for many species, including terrestrial fauna like pangolins (*Manis* spp.) and the Black Rhino (*Diceros bicornis*) and also a number of marine organisms such as sharks and rays (subclass Elasmobranchii) and seahorses (genus *Hippocampus*). China is a leading importer and exporter of many species due to a high demand for luxury products, medicines, and tonics derived from wildlife (*Oldfield, 2003*). One of the most rapidly emerging wildlife trade issues is the demand for, and consumption of, the gill plates of devil and manta rays (mobulids)—marketed under the trade name *Peng Yu Sai*, translated as "Fish Gill of Mobulid Ray." Gill plates—the thin, cartilaginous filaments used to filter plankton and small fish from the water column—are key ingredients in a tonic purported to prevent sickness by boosting the immune system and enhancing blood circulation. The first report of gill plate trade was from the Philippines to China in the 1960s, and international trade rapidly expanded in the late 1990s (*Acebes, 2013*). Traditional Chinese Medicine (TCM) texts first referenced this product in 1976 (*Shen, Jia & Zhou, 2001*), yet recent interviews with practitioners in Guangzhou, China and Singapore stated that *Peng Yu Sai* has no health benefits (*O'Malley et al., 2016*). Furthermore, toxicological studies suggest there are health risks from consuming the high levels of heavy metals in *Peng Yu Sai* (*Wong et al., 2007*; *Li et al., 2012*). It appears that industry marketing of *Peng Yu Sai*, rather than any credible TCM or other medical research, is responsible for its rise in popularity (*Whitcraft, O'Malley & Hilton, 2014*). Regardless of the veracity of the claimed health benefits for humans, this trade poses considerable risk to devil and manta rays.

The life history and ecological traits of mobulids make them highly sensitive to overexploitation. Nine species of devil ray (genus *Mobula*) and two species of manta ray (genus *Manta*) currently make up the subfamily Mobulinae (see examples in Fig. 1). We note, however, that manta rays have been reported to be paraphyletic and nested within

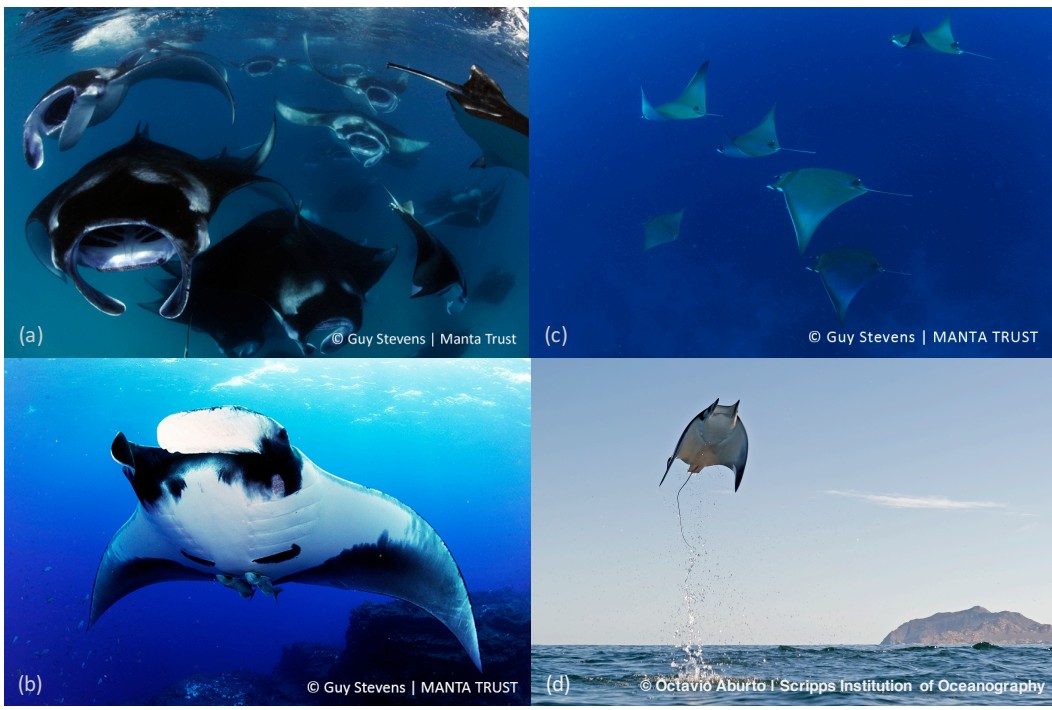

**Figure 1** **Images of devil and manta rays.** (A) Reef Manta Ray (*Manta alfredi*); (B) Oceanic Manta Ray (*Manta birostris*); (C) Shortfin Devil Ray (*Mobula kuhlii*); (D) Smoothtail Devil Ray (*Mobula munkiana*).

the genus *Mobula* and, as such, the taxonomy may change in the near future (*Aschliman, 2014*; *Poortvliet et al., 2015*). Mobulids are filter-feeding planktivores and piscivores that range widely in tropical and warm-temperate waters. The largest devil ray species, the Giant Devil Ray *Mobula mobular* (Bonnaterre, 1788), attains a maximum disc width (analogous to wingspan, as used for bird morphometrics) of five metres (*Notarbartolo di Sciara, 1987*); the largest manta species, the Giant Manta Ray *Manta birostris* (Walbaum, 1792), can reach up to seven metres disc width (*McClain et al., 2015*). Devil and manta rays also have long gestation periods (*Marshall & Bennett, 2010*), and are thought to produce a single pup (*Hoenig & Gruber, 1990*; *Stevens et al., 2000*) every one to three years (*Notarbartolo di Sciara, 1988*; *Compagno & Last, 1999*; *Homma et al., 1999*). Consequently, maximum rates of intrinsic population increase ($r_{max}$; a commonly used metric which reflects the productivity of depleted populations in absence of density dependence), in large mobulid species are among the lowest of all elasmobranchs (*Dulvy et al., 2014*; *Pardo et al., 2016*). Mobulids are taken in a range of targeted fisheries and are also retained or discarded as incidental catch. There is a dearth of species-specific fisheries information because mobulids are often fished and traded under one general category (i.e., all mobulids are landed under the category *manta raya* in Mexican fisheries).

Targeted fisheries for mobulids have existed for decades, yet increased demand for mobulid gill plates has fuelled the emergence and expansion of fisheries targeting these species (*Alava et al., 2002*; *Lewis et al., 2015*). Devil and manta rays were historically exploited for meat (consumed fresh or dried), and to a lesser extent skin (dried) and

cartilage (for shark fin soup filler; *White et al., 2006*; *Acebes, 2013*), and there continues to be a market for some of these devil and manta ray products today. Mobulid meat is used in traditional dishes as a cheap source of protein in Southeast Asia, and in South and Central America (*Croll et al., 2015*). In some countries, such as the Philippines, devil and manta rays are targeted both for their meat, which is consumed domestically, and for their gill plates, which are exported internationally (*Alava et al., 2002*; *Acebes, 2013*). In other countries, such as Sri Lanka, targeted fisheries are driven almost exclusively by the gill plate trade (*Fernando & Stevens, 2011*).

Mobulids are also caught incidentally throughout their ranges (*White et al., 2006*; *Rajapackiam, Gomathy & Jaiganesh, 2007*; *Couturier et al., 2012*), as evidenced by at least 30 large- and small-scale fisheries in 13 countries (*Bonfil & Abdallah, 2004*; *Dapp et al., 2013*; *Croll et al., 2015*). Mobulids are captured in a wide range of gear types including driftnets, purse seines, gillnets, traps, trawls, and longlines (*Croll et al., 2015*). Vessels targeting tuna using gillnets, purse seines, and drift nets are known to capture mobulids, and occasionally retain them as valued catch (*White et al., 2006*; *Fernando & Stevens, 2011*; *Hall & Roman, 2013*). For mobulids caught in and released from purse seine nets, tagging studies indicate moderate-to-high rates of post-release mortality, especially for large individuals that can be difficult to release without physical damage (*Poisson et al., 2014*; *Francis & Jones, 2017*). The handful of studies that have quantified the number of mobulids caught incidentally in certain fisheries and regions suggest that incidental catch may contribute significantly to fishing mortality (*Croll et al., 2015*).

Given the challenges in understanding and regulating mobulid fisheries and trade, conservation planning is necessary to build capacity and ensure comprehensive and collaborative action among stakeholders (*Stanley Price & Soorae, 2003*; *Hoffmann et al., 2008*; *IUCN, 2008*; *Harrison & Dulvy, 2014*). The IUCN began its systematic approach to conservation planning with species-specific Action Plans for large, terrestrial megafauna developed by the Species Survival Commission Species Specialist Groups. These Species Action Plans developed over time to become Species Conservation Strategies, which included a status review as well as a vision, a set of goals, objectives, actions, and targets developed by a group of stakeholders (*IUCN, 2008*). Species conservation planning in the marine realm can be particularly challenging. Compared to terrestrial megafauna, marine megafauna often have larger ranges, making for complicated assessment and abatement of multiple threats across multiple jurisdictions (*McClenachan, Cooper & Dulvy, 2016*). In stark contrast to terrestrial organisms, whose conservation may be significantly advanced by a single country or single organization, marine organisms—especially those where individuals are wide-ranging—require a multi-organization, multi-national approach (*Dulvy et al., 2016*). This paper combines elements from the IUCN Situation Analysis framework—through which major pressures and key regulations are identified (e.g., *Mallon et al., 2015*)—and the IUCN Species Conservation Strategy framework—where a vision, and a set of goals, objectives, and actions are developed.

Concern that international trade controls for manta rays put in place by the Convention on International Trade in Endangered Species of Wild Fauna and Flora (CITES) would create additional pressure on devil rays, combined with the inherent vulnerability of the

entire subfamily, led us to prioritize the development of the Global Devil and Manta Ray Conservation Strategy. A workshop was convened in Durban, South Africa from 9 to 12 June 2014 to develop a Global Devil and Manta Ray Conservation Strategy. In addition to the workshop attendees, inputs from a wider network of devil and manta ray experts informed the Strategy. This paper summarizes of the current state of devil and manta ray scientific research, conservation, and protection (used here to refer to protection obligation, legal or otherwise, and does not examine protection action or effectiveness), and highlights discrepancies between the two genera. We also introduce the global conservation strategy, a living document aimed at guiding future research, policy, and outreach.

## MATERIALS & METHODS

During both the workshop and the development of this document, a significant dichotomy emerged with respect to the understanding of and concern for devil and manta rays. To illustrate this, we first summarize the trends in scientific research on devil and manta rays by updating (a) the systematic literature analysis of *Couturier et al. (2012)* and (b) the maps of species distributions, fisheries, and protection (used here to refer to protection obligation, legal or otherwise, and does not examine protection action or effectiveness). Within this context, we then summarise how the species conservation planning process was undertaken by describing the workshop process and strategy development.

### Scientific research and expertise on devil and manta rays

In order to examine trends in devil and manta ray scientific research, we extracted scientific papers from the ISI Web of Science Core Collection and Google Scholar on 30 May 2016. Our specific aim was to update the systematic literature search conducted up to the year 2011 by *Couturier et al. (2012)* and, as such, we searched the primary literature for titles that contained 'mobula,' 'manta,' or 'devil ray' published from 2012 to 2016. We exported these results and 'false-positive' papers unrelated to devil or manta rays were removed.

### Range, fisheries, and protection mapping

Eleven geographic distribution maps of Extent of Occurrence (EOO) and Area of Occupancy (AOO) were generated prior to the workshop and refined during and following the workshop based on current species distribution knowledge. The EOO is defined as "the area contained within the shortest continuous imaginary boundary which can be drawn to encompass all the known, inferred or projected sites of present occurrence of a taxon" (*IUCN, 2001*; *IUCN, 2012*; *IUCN, 2014*). The AOO is defined as "the area within its 'Extent of Occurrence' that is occupied by a taxon for each country. The measure reflects the fact that a taxon will not usually occur throughout the area of its Extent of Occurrence, which may, for example, contain unsuitable habitats or may be beyond the maximum depth distribution" (*IUCN, 2001*; *IUCN, 2012*; *IUCN, 2014*). The AOO for devil and manta ray species was estimated by including only areas where the presence of a given species had been confirmed.

Devil ray and manta ray AOO distribution maps were grouped by genus in order to map and compare: (a) locations of known fishing pressure through target and incidental

fisheries (from *Croll et al., 2015*), and (b) presence of international, regional, and national protection (detailed in Table 1). Information on known fisheries and protections was gathered primarily by consulting mobulid experts who participated in the conservation strategy workshop, and those who were part of the wider network of experts (detailed below). This information was supported by reviewing the devil and manta ray literature.

### Development of a Global Devil and Manta Ray Conservation Strategy

The IUCN SSG Global Devil and Manta Ray conservation strategy workshop was attended by 18 experts who held knowledge from nine of the 19 Major Fishing Areas as recognized by the Food and Agriculture Organization of the United Nations (FAO; Fig. 2A). Fourteen more experts contributed to the conservation strategy through electronic correspondence, and an additional 16 experts provided knowledge during the 2015 Fisheries Society for the British Isles (FSBI) symposium in Plymouth, United Kingdom (27–31 July 2015; Fig. 2B). These additional collaborators helped to provide expertise for mobulids in the eastern Indian and the Atlantic Ocean, as no workshop participant self-identified as having knowledge specific to these Major Fishing Areas.

Through a series of workshop subgroup discussions and plenary sessions, participants at the strategy workshop developed a vision, goals, objectives, and actions (*IUCN, 2008*) aimed at rebuilding and conserving devil and manta ray populations. This process largely followed that for the Global Sawfish Conservation Strategy, the first of its kind for a group of elasmobranchs (*Harrison & Dulvy, 2014*). The devil and manta ray workshop participants included biologists and fisheries scientists, as well as representatives of organizations focused on tourism, education, and policy.

The group used Specific, Measurable, Achievable, Relevant/Realistic, and Time-Bound (SMART) criteria as a guide for setting objectives and actions. In some cases, workshop participants prioritized countries or regions based on known threats (i.e., those with large, expanding, and unregulated mobulid incidental or targeted catch). Following the workshop, experts revised the goals, objectives, and actions outlined in the strategy, and collaborated on the development of this paper to provide context for the strategy.

## RESULTS

### Scientific research and expertise on devil and manta rays

Since the literature review conducted by *Couturier et al. (2012)*, which included all peer-reviewed literature from 1980 to 2011, the search term "mobula" returned 11 peer-reviewed publications, while "manta" returned 50 over the past 4.5 years from 2012 to 2016 (up to June 2016). The term "devil ray" was also searched for, and returned an additional 10 publications. Compared to *Couturier et al. (2012)*, who reported 96 publications with either "manta" or "mobula" in the title over 31 years (1980–2011), our update identified a total of 71 additional publications with "manta," "mobula," or "devil ray" in the title over only the past 4.5 years. *Couturier et al. (2012)* identified 28 peer-reviewed studies focused on *Mobula* spp. from 1980 to 2011, whereas our study identified 21 novel peer-reviewed studies focused on *Mobula* spp. over the past four years (2012–2016).

Lawson et al. (2017), *PeerJ*, DOI 10.7717/peerj.3027

**Table 1  International, national, and territory/state protections currently in place for devil and manta rays.** International, national, territorial, and state legal protection that restricts fishing and/or trade of a single or multiple species of devil (*Mobula* spp.) and/or manta (*Manta* spp.) ray. The term legal protection is used here to refer to protection obligation, legal or otherwise, and does not examine protection implementation success or effectiveness. The date that this legal protection was passed is included in brackets.

| | *Mobula eregoodootenkee* | *Mobula hypostoma* | *Mobula kuhlii* | *Mobula japanica* | *Mobula mobular* | *Mobula munkiana* | *Mobula rochebrunei* | *Mobula tarapacana* | *Mobula thurstoni* | *Manta alfredi* | *Manta birostris* |
|---|---|---|---|---|---|---|---|---|---|---|---|
| | | | | | International Protections | | | | | | |
| CITES (2016) | ✓ | ✓ | ✓ | ✓ | ✓ | ✓ | ✓ | ✓ | ✓ | | |
| IATTC (2015) | ✓ | ✓ | ✓ | ✓ | ✓ | ✓ | ✓ | ✓ | ✓ | ✓ | ✓ |
| European Union (2015) | ✓ | ✓ | ✓ | ✓ | ✓ | ✓ | ✓ | ✓ | ✓ | ✓ | |
| GFCM (2015) | | | | | ✓ | | | | | | |
| CMS Appendix I & II (2014) | ✓ | ✓ | ✓ | ✓ | ✓ | ✓ | ✓ | ✓ | ✓ | ✓ | |
| CITES Appendix II (2013) | | | | | | | | | | ✓ | ✓ |
| European Union (2012) | | | | | | | | | | | ✓ |
| CMS Appendix I & II (2011) | | | | | | | | | | | ✓ |
| Barcelona Convention SPA/BD Protocol Annex II (2001) | | | | | ✓ | | | | | | |
| Bern Convention Appendix II (2001) | | | | | ✓ | | | | | | |

Lawson et al. (2017), *PeerJ*, DOI 10.7717/peerj.3027

**Table 1** (*continued*)

| | Mobula eregoodootenkee | Mobula hypostoma | Mobula kuhlii | Mobula japanica | Mobula mobular | Mobula munkiana | Mobula rochebrunei | Mobula tarapacana | Mobula thurstoni | Manta alfredi | Manta birostris |
|---|---|---|---|---|---|---|---|---|---|---|---|
| National Protections | | | | | | | | | | | |
| Peru (2016) | | | | | | | | | | | ✓ |
| Australia (2015) | ✓ | ✓ | ✓ | ✓ | ✓ | ✓ | ✓ | ✓ | ✓ | ✓ | |
| Indonesia (2014) | | | | | | | | | | ✓ | ✓ |
| Maldives (2014) | ✓ | ✓ | ✓ | ✓ | ✓ | ✓ | ✓ | ✓ | ✓ | ✓ | ✓ |
| United Arab Emirates (2014) | | | | | | | | | | ✓ | ✓ |
| Brazil (2013) | ✓ | ✓ | ✓ | ✓ | ✓ | ✓ | ✓ | ✓ | ✓ | ✓ | ✓ |
| Australia (2012) | | | | | | | | | | | ✓ |
| Ecuador (2010) | | | | ✓ | | ✓ | | ✓ | ✓ | | ✓ |
| New Zealand (2010) | | | | ✓ | | | | | | | ✓ |
| Mexico (2007) | | ✓ | | ✓ | | ✓ | | ✓ | ✓ | | ✓ |
| Croatia (2006) | | | | | ✓ | | | | | | |
| Israel (2005) | ✓ | ✓ | ✓ | ✓ | ✓ | ✓ | ✓ | ✓ | ✓ | ✓ | ✓ |
| Malta (1999) | | | | | ✓ | | | | | | |
| Philippines (1998) | | | | | | | | | | | ✓ |
| Territory and State Protections | | | | | | | | | | | |
| West Manggarai/Komodo, Indonesia Regency (2013) | | | | | | | | | | ✓ | ✓ |
| Raja Ampat, Indonesia Regency (2012) | ✓ | ✓ | ✓ | ✓ | ✓ | ✓ | ✓ | ✓ | ✓ | ✓ | ✓ |
| Guam, USA Territory (2011) | ✓ | ✓ | ✓ | ✓ | ✓ | ✓ | ✓ | ✓ | ✓ | ✓ | ✓ |

**Table 1** (*continued*)

| | Mobula eregoodootenkee | Mobula hypostoma | Mobula kuhlii | Mobula japanica | Mobula mobular | Mobula munkiana | Mobula rochebrunei | Mobula tarapacana | Mobula thurstoni | Manta alfredi | Manta birostris |
|---|---|---|---|---|---|---|---|---|---|---|---|
| Christmas Island and Cocos (Keeling) Islands, Australian Indian Ocean Territories (2010) | | | | | | | | | | ✓ | ✓ |
| Hawaii, USA State (2009)[a] | | | | | | | | | | ✓ | ✓ |
| Yap, Federated States of Micronesia (2008) | | | | | | | | | | ✓ | ✓ |
| Commonwealth of the Northern Mariana Islands, USA Territory (2007) | ✓ | ✓ | ✓ | ✓ | ✓ | ✓ | ✓ | ✓ | ✓ | ✓ | ✓ |
| Florida, USA State (2006) | ✓ | ✓ | ✓ | ✓ | ✓ | ✓ | ✓ | ✓ | ✓ | ✓ | ✓ |

**Notes.**
[a] A bill is currently under consideration by Hawaii's state legislature to expand protection to include all sharks and rays.
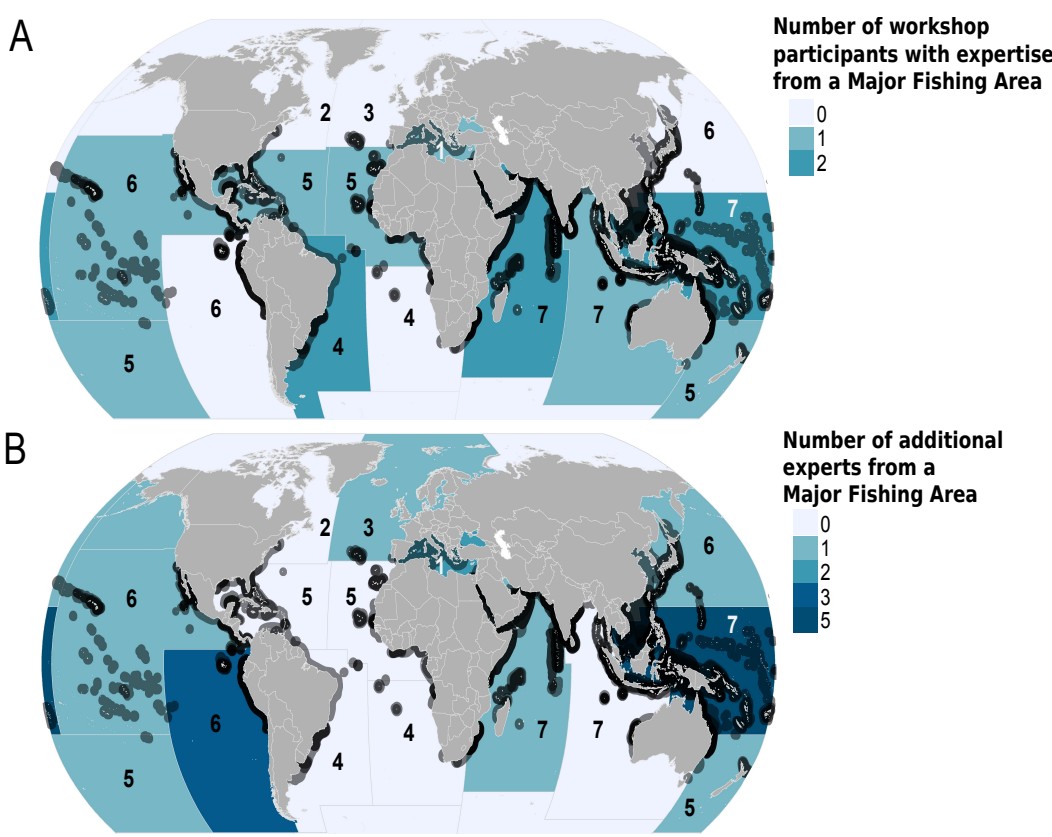

**Figure 2** **Geographic extent of the expertise that contributed to the Global Devil and Manta Ray Conservation Strategy.** Geographic extent of the expertise is shown using Food and Agriculture Major Fishing Areas (FAO MFA) and species Area of Occupancy (AOO) maps. The dark grey outlines around the countries indicate the presence of one or more mobulid AOO, and the number within each FAO MFA represents the number of species per major fishing area. The degree of colour saturation in each FAO MFA represents the number of experts (A) who attended the workshop, and (B) additional experts who shared information via electronic correspondence and/or from during the Fisheries Society of the British Isles symposium.

For devil rays (21 studies), the research theme (as defined by *Couturier et al., 2012*) was dominated by "Occurrence" (10); followed by "Anatomy, biology, and morphology" (4), and "Bycatch and fisheries" (3). For manta rays (44 studies), the top two research themes were also "Occurrence" (15) and "Anatomy, biology, and morphology" (13), followed by "Life history and population" (7) and "Bycatch and fisheries" (4). Publications that referred to both devil and manta rays in the title (6 studies) were excluded in this research theme analysis. These results differ from *Couturier et al. (2012)*'s finding that across all devil and manta ray studies "Taxonomy" was the leading research theme, followed by "Occurrence" and "Bycatch and fisheries."

## Range, fisheries, and protection mapping

Novel reports on occurrence of mobulid species updated species-specific Area of Occupancy and Extent of Occurrence maps for the eleven species of devil and manta ray (Fig. 3). Compared to the previous IUCN Red List distribution maps (*IUCN, 2015*), these updated

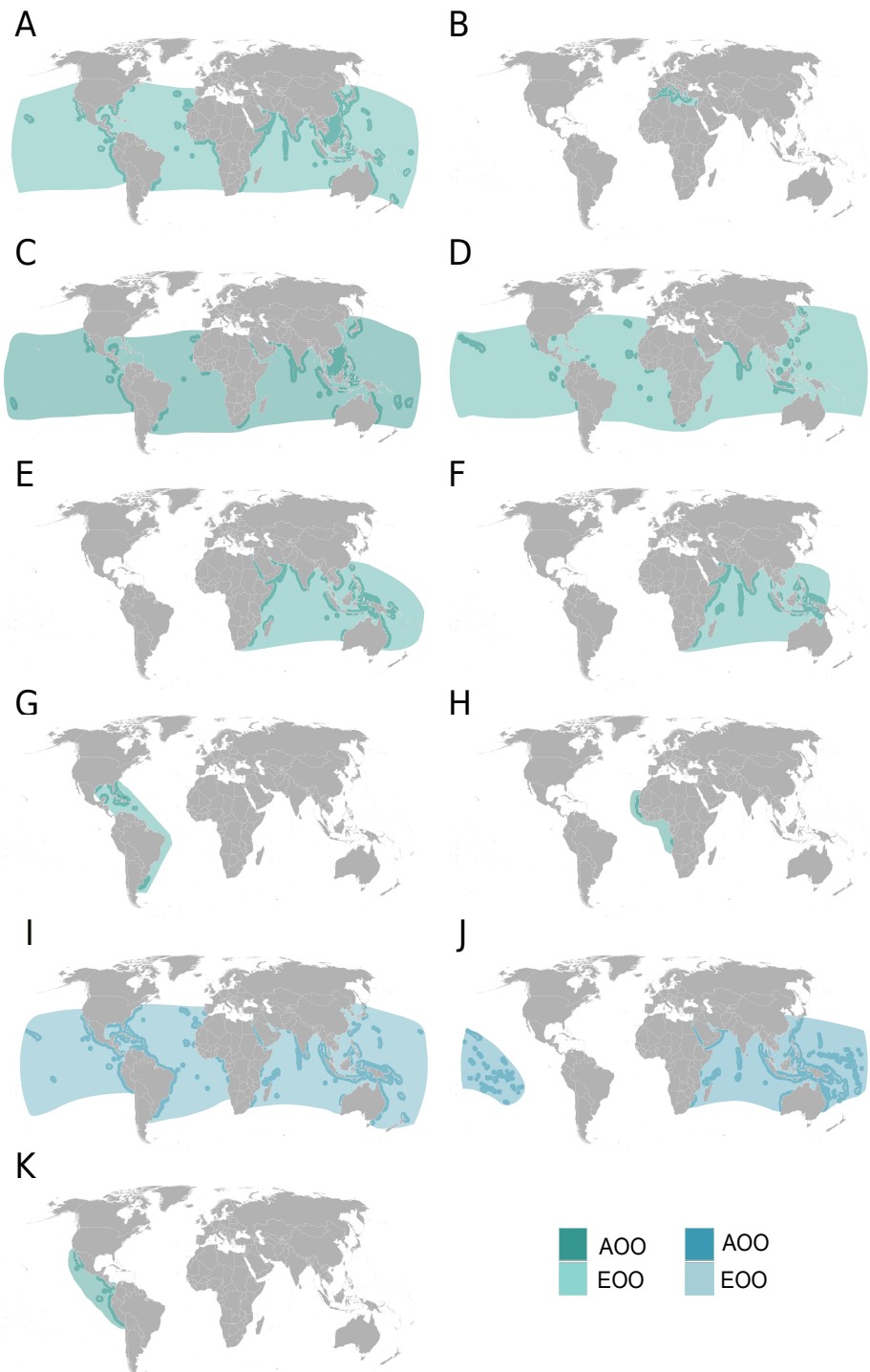

**Figure 3  Distribution maps for manta and devil ray species.** Extent of Occurrence (EOO) and Area of Occupancy (AOO) maps for all nine species of devil ray and both species of manta ray. Species are as follows: (A) *Mobula japanica*; (B) *Mobula mobular*; (C) *Mobula thurstoni*; (D) *Mobula tarapacana*; (E) *Mobula eregoodootenkee*; (F) *Mobula kuhlii*; (G) *Mobula hypostoma*; (H) *Mobula rochebrunei*; (I) *Manta birostris*; (J) *Manta alfredi*; (K) *Mobula munkiana*.

maps show larger AOO and EOOs for most mobulids, as new sightings and landings data have been reported. For example, the Chilean Devil Ray *Mobula tarapacana* (Philippi, 1892) is confirmed to be present throughout the Red Sea and Gulf of Aden, whereas previously it was only reported in the upper Red Sea. Generally, there have been range extensions across the central Atlantic with new sightings of the Bentfin Devil Ray *Mobula thurstoni* (Lloyd, 1908) in West Africa, ranging across the isolated Atlantic islands of the Canaries, Azores, and Ascension. Furthermore, there were significant new records in the Indo-west Pacific nations of the Philippines, Vanuatu (for the Bentfin Devil Ray), Fiji and Palau (for the Chilean Devil Ray).

The devil rays fall within three broad classes of geographical distribution. The three largest-bodied species—at least 1.8 m disc width—have near circumglobal tropical and subtropical geographic ranges: the Spinetail Devil Ray *Mobula japanica* (Müller & Henle, 1841), Chilean Devil Ray, and Bentfin Devil Ray. Three species are found only in the Atlantic Ocean and Mediterranean Sea. The Atlantic Devil Ray *Mobula hypostoma* (Bancroft, 1831) is found only in the western Atlantic and has an apparently disjunct distribution in the Caribbean and northern Gulf of Mexico and also in the South Atlantic continuous along the coastlines of Southern Brazil, Uruguay, and Northern Argentina. The eastern Atlantic counterpart is the Lesser Guinean Devil Ray *Mobula rochebrunei* (Vaillant, 1879) that was described from Guinea and is thought to be present in Mauritania, Senegal, Guinea-Bissau, and Angola. Finally, the largest species—the Giant Devil Ray—is apparently only found in the Mediterranean Sea. There are three smaller (approximately 1 m disc width) Indo-Pacific species: two found only in the Indian Ocean and western Pacific Ocean, the Pygmy Devil Ray *Mobula eregoodootenkee* (Bleeker, 1859) and the Shortfin Devil Ray *Mobula kuhlii* (Müller & Henle, 1841), and one species found only in the eastern Pacific, the Smoothtail Devil Ray *Mobula munkiana* Notarbartolo di Sciara, 1987. The larger of the two manta ray species, the Giant Manta Ray, has a near circumglobal distribution in tropical and subtropical waters and is most similar to those of the three largest devil ray species, whereas the Reef Manta Ray *Manta alfredi* (Krefft, 1868) has an Indo-West Pacific distribution.

Species-specific differences in international, national, and state/territory protection occur across the 11 mobulid species. The Giant Manta Ray has the largest number of international, national, and state/territory protection of any mobulid species, followed by the Reef Manta Ray (Table 1). Species-specific differences exist within the devil rays. The Giant Devil Ray has a small EOO that coincides with the Mediterranean Sea (Fig. 3B) yet it has been the subject of numerous national, regional, and international protection commitments from surrounding countries (Table 1). In contrast, the Chilean Devil Ray has a large EOO that overlaps with that of the Giant Manta Ray, but national protection is only afforded in six of the 31 countries in its recorded range (Table 1).

When grouped by genus, protections for devil rays under two key international agreements are now equal to those for manta rays (Fig. 4), yet national and state/territory legislation for devil rays still lags behind protection for manta rays (Fig. 5). Moreover, current national regulations leave devil rays unprotected and manta rays protected in areas where target and incidental fisheries for both genera are known to occur. For example, several mobulid target and incidental fisheries occur in the Indo-Pacific region and most

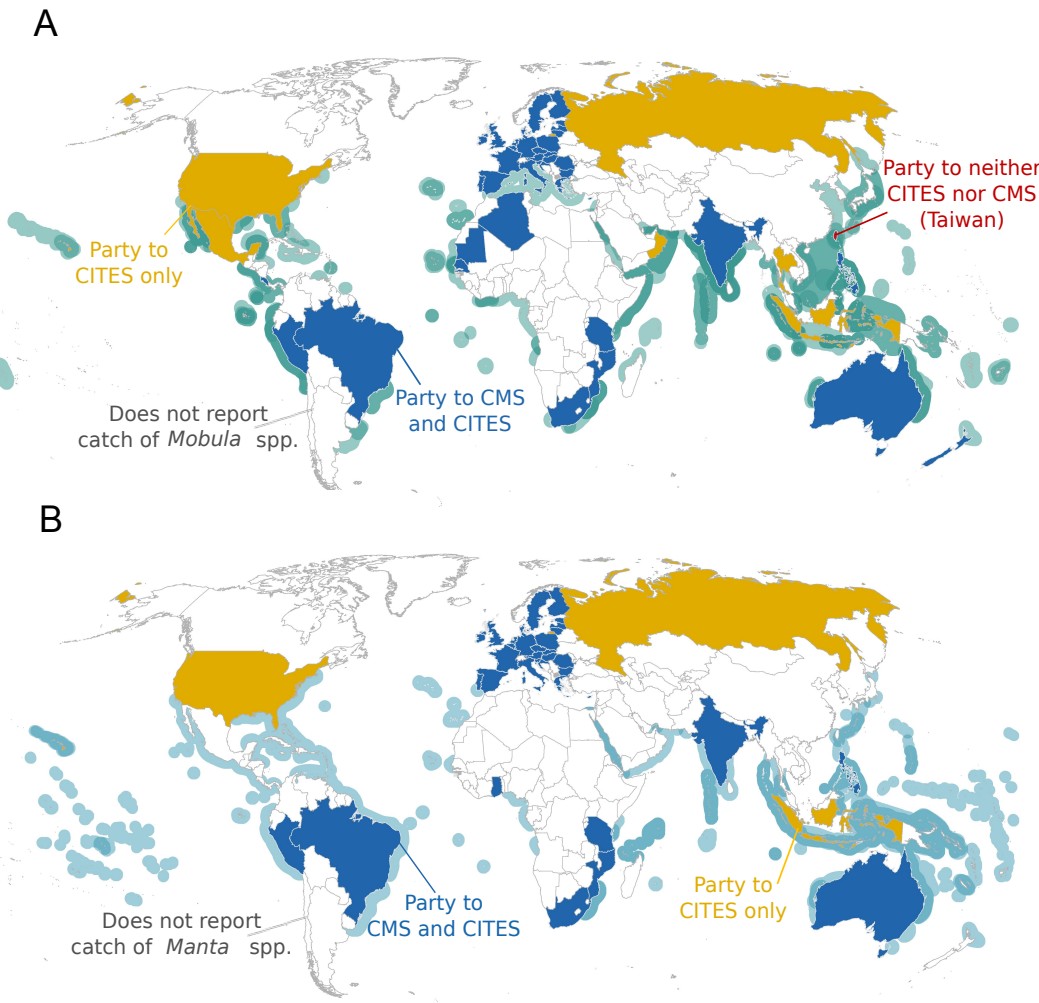

**Figure 4** **Distribution of Parties to CITES and CMS with respect to ranges of mobulid species.** Country participation in two key international protection agreements as it relates to each genus; the Convention on International Trade in Endangered Species of Wild Fauna and Flora (CITES) and the Convention on the Conservation of Migratory Species of Wild Animals (CMS) with respect to Area of Occupancy maps for a single or multiple species of (A) devil (*Mobula* spp.) or (B) manta (*Manta* spp.) ray. Only countries that are known to obtain mobulids in target or incidental fisheries are included, whereas those that do not report target or incidental fisheries for mobulids are blank (see *Croll et al., 2015* for details). Both nearshore and distantwater fleets are included, thus country of origin may not overlap with mobulid distribution if fisheries operate elsewhere.

protections in this region apply exclusively to manta rays (i.e., Indonesia, Peru, and the Philippines; see Fig. 5).

## Development of a Global Devil and Manta Ray Conservation Strategy

Workshop participants agreed on an overall vision for the status of devil and manta rays, and three goals aimed at achieving this vision, as well as a series of sixteen objectives and associated actions to support the goals (see Table 2 for details)

**Vision:** *Populations of devil and manta rays that flourish in resilient ocean ecosystems, harmoniously with human communities, through knowledge, sustainability, and education.*

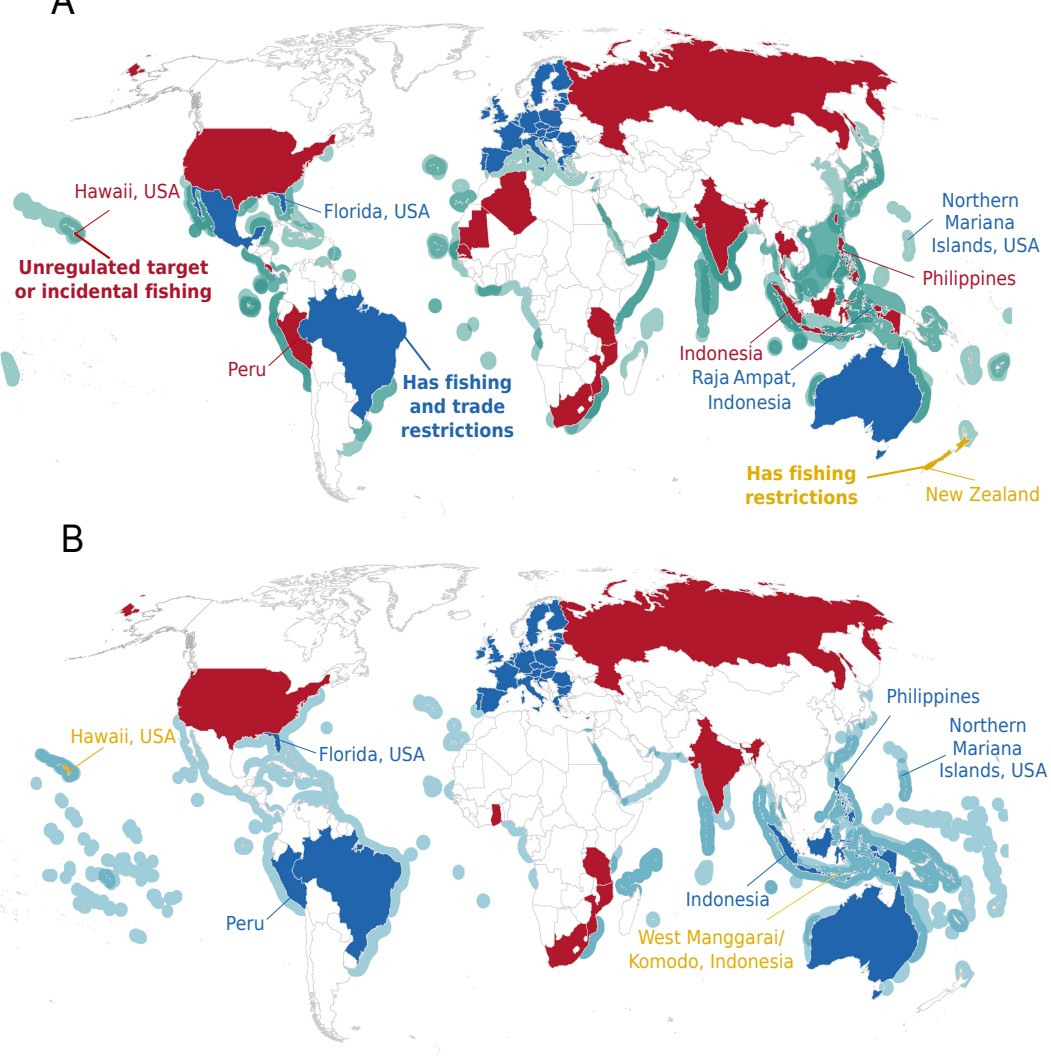

**Figure 5** **Distribution of national, territory, and state protections with respect to ranges of mobulid species.** National, state, and territorial legislation that restricts fishing and/or trade (excludes international obligations), with respect to Area of Occupancy maps for a single or multiple species of (A) devil (*Mobula* spp.) and (B) manta (*Manta* spp.) ray. Only countries that are known to obtain mobulids in target or incidental fisheries were included, whereas those that do not report target or incidental fisheries for mobulids are blank (see *Croll et al., 2015* for details). Both nearshore and distant-water fleets are included, thus country of origin may not overlap with mobulid distribution if fisheries operate elsewhere. Note that the countries of the European Union are grouped and included in this map to be consistent with *Croll et al. (2015)*.

**Goal A.** The knowledge required to sustain devil and manta rays is generated and communicated.

**Objectives:**

1. Taxonomy: To resolve the taxonomy and define management units of devil and manta rays

**Table 2  The Global Devil and Manta Ray Conservation Strategy.** The complete text of the Global Devil and Manta Ray Conservation Strategy; including a vision, and a series of goals, objectives, and actions.

**Vision:** *Populations of devil and manta rays that flourish in resilient ocean ecosystems, harmoniously with human communities, through knowledge, sustainability, and education*

**Goal A: The knowledge required to sustain devil and manta rays is generated and communicated to relevant stakeholders.**

**Objective 1**

**Taxonomy and stock structure**

Taxonomy of devil and manta rays is resolved, and management units are defined.

The taxonomy of devil and manta rays is still unclear and substantial changes at the species and even genus level are expected. Defining management units will enable more focused and efficient conservation measures for these species, and show where trans-national regulations are necessary.

**Actions**

1.1 Produce peer-reviewed publications that resolve the species-level taxonomy of devil and manta rays to be used by the scientific and management community.

1.2 Undertake research to define management units of devil and manta ray populations on regional and global scales.

1.3 Refine a list of priority species and regions based on newly defined management units.

    1.3.1 Potential priority species include *M. japanica*, *M. tarapacana*, *M. mobular*.

    1.3.2 Potential priority regions include the Indo-Pacific, Mediterranean Sea, Eastern Pacific, and West Africa.

**Objective 2**

**Biology**

Productivity, life history, and demography of devil and manta rays are determined.

Information describing biological characteristics, such as annual fecundity and age at maturity are needed to fully understand the vulnerability of these species and enable prioritization of conservation and management actions.

**Actions**

2.1 Produce a standardized data collection methodology and a guide to facilitate mobulid biology data comparison among research groups and countries.

2.2 Define accurate biological parameters (age, growth, maximum age, and age at maturity data) for devil and manta ray populations for use in species assessments, scientific reports, and publications.

2.3 Use population data to determine the rate of natural mortality in devil and manta ray populations for integration into species assessments.

**Objective 3**

**Ecology**

Spatial and temporal ecology of devil and manta rays is understood.

Ecological data are needed to inform appropriate management actions that prevent overexploitation of devil and manta rays, preserve connectivity among populations, and protect critical habitats.

**Actions**

3.1 Consolidate and synthesize available data to determine historic and core distributions of mobulid species, in order to aid recovery and assess potential reestablishment throughout historic ranges.

3.2 Update Extent of Occurrence and point distribution maps of the geographic distribution of devil and manta rays and disseminate this information.

3.3 Describe and define areas of critical habitat and population connectivity (by size, sex and reproductive status) including areas of core use (hot spots, aggregation sites), seasonality of presence, and migratory corridors to produce high resolution geographic outputs for publication and management actions (e.g., place-based protection).

3.4 Understand the role that diet and feeding ecology have in predicting aggregations, movement, and habitat use of devil and manta rays.

3.5 Estimate the abundance of devil and manta ray species using information collected by fisheries-independent research programs (e.g., line transect surveys, photo identification, tagging).

**Table 2** (*continued*)

**Objective 4**
**Strategic research and communication**

Scientific research on biology, ecology, status, threats and socio-economic value of devil and manta rays to enhance conservation and management is communicated to stakeholders and to the public.

Building an improved understanding of the status and threats that face devil and manta rays among the general public, policy makers and the conservation and management community is helpful for the implementation of national and international conservation legislation, and will engage the public to support protecting these species.

**Actions**

4.1 Update International Union for the Conservation of Nature (IUCN) Red List global re-assessments for mobulid species.

    4.1.1 Priority species include *Mobula japanica*, *M. tarapacana*, and *M. thurstoni*.

4.2 Produce a global status summary of devil and manta ray fisheries and catches.

4.3 Translate research for the wider conservation and management community including Non-Governmental Organizations (NGOs) as well as fishers, tourism, divers, aquarists, etc. through newsletters, social, print, and traditional media outlets, as information becomes available.

4.4 Interpret research for managers and policy-makers to help inform decisions related to the protection and conservation of devil and manta ray populations as opportunities arise at key management decision points (such as CITES, CMS, Convention for Biological Diversity, RFMO meetings, local management meetings, and national biodiversity initiatives).

**Goal B: Devil and manta ray populations are maintained at, or recovered to, ecologically relevant levels by managing fisheries, trade, and demand.**
**Objective 5**
**Fisheries assessment and management**

Devil and manta ray populations and fisheries in which they are taken are monitored and managed for long-term sustainability.

Unmanaged and mostly unmonitored fisheries pose the greatest threat to devil and manta rays. Standardized data collection is needed to assess population trends and inform conservation measures to prevent overexploitation from targeted and incidental mortality.

**Actions**

5.1 Create incentives for government policy makers to take action on devil and manta ray conservation and management through positive international media opportunities.

5.2 Collate historical landings and market data.

5.3 Develop standardized guidelines for fisheries data collection (e.g., species identification and sizing, tissue samples, reproductive status) and monitoring (e.g., landings, discards, fishing effort, gear types).

    5.3.1 Develop observer practices that are specific to devil and manta rays (e.g., tissue samples, reproductive data, size estimation, etc.).

    5.3.2 Develop a multilingual identification guide/webpage/app to assist observers/customs officers/scientists/NGOs in identification, data collection, etc.

5.4 Adopt a standardized data collection system across national, state, and/or regional fisheries departments that gathers information on landings, bycatch, and discards using at-sea and landing site observer programs.

5.5 Report national species-specific landings of devil and manta rays to FAO and/or RFMOs.

5.6 Determine areas of overlap between devil and manta ray distributions and relevant fisheries to identify priority areas to minimize bycatch.

5.7 Estimate the total annual volume of devil and manta ray catch in fisheries bycatch globally, by region, and by gear type.

5.8 Develop gears and fishing practices that minimize bycatch.

    5.8.1 Review handling and release procedures using different gears and develop and implement best practice procedures where they don't exist.

    5.8.2 Produce education and outreach materials about safe release and handling.

    5.8.3 Reduce purse seine sets in locations, during times of year, and in set types where mobulids have been identified as bycatch.

5.9 Estimate post-release mortality across various sizes, species, and gear types for devil and manta rays.

5.10 Develop stock assessment methods for devil and manta rays and coordinate the appropriate agencies, NGOs, and/or fisheries scientists to undertake assessments.

**Table 2** (*continued*)

5.11 Identify and prioritize species and stocks that require assessment within each RFMO, region, and nation.

5.12 Regularly assess and report the status of devil and manta ray fisheries and estimate sustainable catch levels in each RFMO, region, and nation.

5.13 Implement and enforce protections for devil and manta rays to maintain or recover stocks to ecologically relevant levels in each RFMO, region, and nation.

5.14 Harmonize management arrangements between adjacent nations to ensure consistent assessment of shared stocks and to coordinate data collection.

5.15 Ensure that important devil and manta ray aggregation sites are protected through existing and/or revised spatial and temporal management measures in each RFMO, region, and nation.

**Objective 6**
**Trade regulation**

Imports and exports of devil and manta ray products are traceable, monitored, and regulated for sustainability.

Manta rays were listed under CITES Appendix II in 2013, and devil rays in 2016, meaning that CITES Parties are obliged to monitor and regulate international imports and exports of manta parts, including gill plates. Supporting efforts to monitor and regulate trade is critical to identifying sources of demand and supply and preventing unsustainable levels of trade.

**Actions**

6.1 Enforce and implement legislation of international conservation agreements for devil and manta rays (e.g., CITES, CMS, and RFMOs).

6.2 Develop and disseminate identification guides for traded devil and manta ray products.

6.3 Ensure the adoption of customs codes for (a) CITES-listed species, and (b) gill plate products.

6.4 Develop a CITES Non-Detriment Finding (NDF) guide to support the implementation of CITES listings in key devil and manta ray fishing nations.

6.5 Produce country-of-origin standardized certificates for all gill plate exporting and importing states.

6.6 Implement port-state controls (the inspection of foreign vessels by official officers) for all range states.

6.7 Provide catch documentation for individual consignment of gill plates by issuing authorities.

6.8 Conduct market surveys at regular intervals.

6.9 Compare and confirm market survey data with trade data reported by exporters and importers.

6.10 Propose *Mobula* spp. for inclusion on Appendix II of CITES in collaboration with NGOs, scientists, and devil and manta ray range states.

**Objective 7**
**Socio-economics and markets**

Demand for devil and manta ray products is reduced, and an understanding of socio-economic drivers is informing management.

Demand for devil and manta ray gill plates (*Peng Yu Sai*) has been cited as the leading driver of increased directed fisheries since the late 1990s. Reducing demand for devil and manta ray gill plates and other products including meat, cartilage, and skin will remove a strong economic incentive that is driving overexploitation of these species.

**Actions**

7.1 Understand the socio-economic value and landscape of consumptive uses of devil and manta rays.

7.2 Understand the socio-economic value and landscape of non-consumptive uses of devil and manta rays.

7.3 Assess the current demand for *Peng Yu Sai* and the level of consumer awareness to the threats posed by the gill plate market.

7.4 Produce a profile of the typical consumer of *Peng Yu Sai* in order to most effectively and efficiently target the demand reduction campaign.

7.5 Determine what the current marketing channels and methods for promoting use of *Peng Yu Sai* are.

    7.5.1 Determine the extent of TCM practitioner involvement in recommending or marketing *Peng Yu Sai* and the opinions and attitudes of TCM practitioners regarding *Peng Yu Sai* use and efficacy.

7.6 Update 2011 assessment of *Peng Yu Sai* markets in Guangzhou, China, by collecting samples, conducting and analysing toxicology tests, and producing a report summarizing assessment results.

7.7. Produce material, media, and social media and recruit spokespeople and media partners to join a campaign that draws attention to threats posed by the gill plate market.

**Table 2** (*continued*)

7.8 Conduct a follow-up assessment both directly and by third parties to measure effectiveness of the campaign using qualitative (changes in attitudes, level of awareness) and quantitative measures (evidence of reduced consumption, reduction in gill plate sales), measured against a baseline assessment.

    7.8.1 Ensure ongoing monitoring of the distribution of Public Service Announcements, short films, and earned media across a variety of media delivery platforms, measured in economic value and target audiences reached.

    7.8.2 Communicate with media sources for feedback regarding changes in *Peng Yu Sai* demand and trade.

    7.8.3 Communicate with partners and collaborators engaged in monitoring key devil and manta ray landing sites in Indonesia and Sri Lanka for feedback regarding changes in mobulid landings, and reported changes in demand or prices from gill plate traders.

**Goal C: Educated and engaged communities are supporting and benefiting from devil and manta ray conservation and management through improved livelihoods.**
**Objective 8**
**Tourism**

A standardized best practice approach to tourism interactions with devil and manta rays that minimizes harm is adopted and enforced by tourism operators globally.

Non-consumptive use of devil and manta rays through responsibly managed tourism can provide long-term sustainable economic benefits to coastal communities as one alternative to unsustainable fisheries. Standardized best practice guidelines for tourism operators will prevent injury and stress to the animals and environments, while making the businesses that rely on healthy devil and manta ray populations more environmentally sustainable, and ultimately, more successful.

**Actions**

8.1 Collate and standardize the existing best practices of devil and manta ray tourism interactions (e.g., diving, snorkelling, and watching).

8.2 Develop best practice guidelines for tourism interactions with devil and manta rays.

8.3 Secure adoption of best practice guidelines for tourism interactions with devil and manta rays by the wider tourism community.

8.5 Educate snorkelers as well as recreational and professional SCUBA divers about the conservation and management of devil and manta rays through development and dissemination of offline and online educational tools including specialty training.

**Objective 9**
**Community engagement**

Knowledgeable communities are contributing to devil and manta ray conservation and management at the local level.

Communicating the benefits of devil and manta ray conservation and including community stakeholders in the process is essential to adoption, implementation, and enforcement of conservation and management measures.

**Actions**

9.1 Produce and distribute engaging and compelling media to inspire the general public in key fishing countries and globally to support devil and manta ray conservation measures.

9.2 Engage indigenous and local fishing communities in sharing of traditional ecological knowledge and cultural value (e.g., animal totems) of historical species composition, species distribution and temporal occurrence.

9.3 Create and deliver road shows, stage shows, or film events to highlight the conservation status of devil and manta rays in coastal fishing communities that are adjacent to devil and manta ray populations in priority countries (e.g., Philippines, Indonesia, Sri Lanka, and Peru).

9.4 Create interpretive material to communicate the value of devil and manta rays tourism through social media, websites, magazines, print, and television to the government, local communities, and global supporters of NGOs.

9.5 Engage tourism operators and the public to report sightings by submitting ventral photographs to an online identification database.

9.6 Translate a global identification guide for devil and manta rays into the local languages of the priority fishing nations (e.g., Peru, Philippines Indonesia, India, Mexico, and Sri Lanka).

**Objective 10**
**Alternative livelihoods**

People in coastal communities are engaging in occupations and subsistence activities that are not based on exploitation of devil and manta rays.

Empowering coastal communities to transition away from dependence on unsustainable fishing practices and into alternative livelihoods (e.g., sustainable fisheries, aquaculture, and tourism) is essential to the success of devil and manta ray conservation and management measures and the economic future of the communities.

Actions

10.1 Consult and work with social and climate scientists, and development agencies to identify opportunities for the development of alternative livelihoods for coastal fishing communities and work to ensure that the conservation of devil and manta rays is included in their objectives.

10.2 Identify potential markets for developing ecotourism-based alternative livelihoods in local government (e.g., tourism board and development assistance), and in sustainable tourism businesses (e.g., hotels).

10.3 Develop alternative livelihoods and income opportunities for at least five local communities in at least five of the main devil and manta ray fishing nations (e.g., Peru, Philippines Indonesia, India, Mexico, and Sri Lanka) to diversify away from fishing for devil and manta rays.

10.4 Build capacity in local communities and among artisanal fishermen through training (business, tourism management, and sustainable fishing and aquaculture practices) and assistance with raising capital for the expenses associated with implementation.

**Objective 11**
**Devil and manta ray network**

Devil and manta ray experts support government and private sector bodies by encouraging, prioritizing, facilitating, integrating, and fulfilling commitments to conservation plans and regulations.

The devil and manta ray network provides an important forum for sharing and propagating conservation knowledge, generating coordinating actions, and monitoring progress.

Actions

11.1 Conduct at least one workshop for representatives of government, policy makers, and trade officials in each priority fisheries country (e.g., Peru, Philippines, Indonesia, India, Mexico, Sri Lanka, and the Gaza Strip) on the conservation status and state of devil and manta ray international trade and provide training in the identification of gill plates and species.

11.2 Connect NGOs and fishing organizations with interested scientists to develop, fund, and implement collaborative projects aimed at gaining government buy-in and building government champions.

    11.2.1 Form of a coalition of contributors united toward devil and manta ray conservation with different areas of expertise (e.g., science, policy, media, community outreach) from different regions.

    11.2.2 Identify and develop opportunities for collaborative, resource-effective, research and conservation programs (e.g., IUCN Specialist Groups, NGOs) with other aquatic vertebrates that share habitat and threats with devil and manta rays (e.g., cetaceans, whale sharks and other elasmobranchs).

    11.2.4 Coordinate comments, speaking opportunities, and advocacy around key government decision meetings.

11.3 Commit to ongoing engagement by NGOs and scientists to articulate and promote devil and manta ray conservation plan goals to governments.

    11.3.1 Engage in regular contact and discussion with key government officials.

    11.3.2 Attend national and/or RFMO science, bycatch, and/or ecosystem committee meetings.

    11.3.3 Prepare written comments to national fisheries and/or environment government leads and/or RFMO chairs.

    11.3.4 Serve on government delegations to key decision meetings including CITES and CMS Conferences of Parties and RFMO annual meetings.

    11.3.5 Participate in targeted side events at key meetings to bring together various interests toward a common goal.

11.4 IUCN SSG and partners review progress and revise actions under the Global Devil and Manta Ray Conservation Strategy every three years.

11.5 Ensure a continued stream of financial resources to ensure timely implementation of the Actions included in this Global Devil and Manta Ray Conservation Strategy.

2. Biology: To determine the productivity, life history, and demography of devil and manta ray populations
3. Ecology: To understand the spatial and temporal ecology of devil and manta rays
4. Public Communication: Communicate research on biology, ecology, status, threats and socio-economic value, of devil and manta rays to enhance conservation and management.

**Goal B.** Devil and manta ray populations are maintained at, or recovered to, ecologically relevant levels through managing fisheries, trade and demand.

**Objectives:**
5. Fisheries Management: Devil and manta ray populations and fisheries are monitored and sustainably managed
6. Trade Regulation: Ensuring that trade in all devil and manta ray products is traceable, regulated, and monitored
7. Socio-economics and Demand: Reduce the demand for devil and manta ray products and understand the socio-economic drivers to ensure demand does not drive unsustainable fishing.

**Goal C.** Educated and engaged communities support and benefit from devil and manta ray conservation and management through outreach, capacity building, and fundraising.

**Objectives:**
8. Tourism: A standardized best practice approach to tourism interaction with devil and manta rays is adopted by tourism operators globally
9. Community Engagement: Knowledgeable communities contribute to devil and manta ray conservation and management
10. Alternative Livelihoods: Empowered coastal communities benefit from alternative livelihoods that are developed to reduce overexploitation of devil and manta ray populations
11. Devil and Manta Ray Network: Commitments to plans and regulations are encouraged, prioritized, facilitated and fulfilled.

## DISCUSSION

There has been a recent quantum leap in scientific research, conservation campaigns, and policy directives aimed at understanding and conserving devil and manta rays. Attention to lesser-known devil rays, however, has clearly lagged behind such initiatives for well-known manta rays. We draw on well-established frameworks for species conservation planning that inoculate against implicit or idiosyncratic values. Using such an approach, we have articulated a first-draft conservation strategy with a common vision and goals for all mobulids. To provide context for the conservation strategy and address potential roadblocks to its success, we (1) examine how this apparent charisma gap between devil and manta rays arose, (2) track the recent path of manta ray protection to see where this protection can be extended to include the devil rays, (3) consider how responsible trade and demand reduction can curtail targeted fishing, and (4) and examine how incidental fishing mortality can be minimized.

## How did the charisma gap arise between manta and devil rays?

Tourism and related economic sectors have partially fuelled the conservation activities surrounding manta rays. Translating this success to devil rays, however, may be challenging because of biological differences that make devil rays harder for tourists to easily and reliably access. Both species of manta ray reach a large body size, form predictable aggregations, and are accessible to divers (*O'Malley, Lee-Brooks & Medd, 2013*). Some species of devil ray also reach a large body size, form predictable aggregations, and are accessible to divers, and as a result may be incorrectly identified as manta rays by tourism operators and tourists (RHL Walls & D Fernando, pers. obs., 2016). Other devil ray species, however, form unpredictable and sporadic aggregations and exhibit shy behaviour. For example, while aggregations of leaping Smoothtail Devil Ray (Fig. 1D) off Baja California provide great potential for boat-based tourism, they occur over short periods that can be difficult to predict. It is inevitable that sighting frequency and reliability will enable a larger segment of society to engage with manta rays, resulting in greater overall interest in conserving manta rays than for devil rays, despite similar threats across genera.

This is not the first charisma gap in species conservation, nor chondrichthyan conservation, and this pattern may be more widespread than is appreciated (*McClenachan et al., 2012*). For example, an apparent charisma gap occurs in the United States where the 2010 Shark Conservation Act prohibited removal of fins at sea for all sharks landed in United States waters, with an exception for the Atlantic Dusky Smooth-hound (*Mustelus canis*), which can be landed with their fins removed according to an exceptionally lenient fin-to-carcass ratio (*United States Public Law 111–348, 2011*). US Atlantic state bans on shark fins also make exceptions for smooth-hounds as well as the Spiny Dogfish (*Squalus acanthias*). In many places around the world, shark finning has been banned largely based on concerns over cruelty, yet few regulations prohibit the removal of wings from live skates. We caution that variation in public awareness, and care for a biased subset of chondrichthyans and other marine organisms, will lead to problematic asymmetries in scientific knowledge, conservation campaigns, and protective regulations (*McClenachan et al., 2012*). We suggest using well-established species conservation planning frameworks for a means of bridging the charisma gap.

## How can we build upon manta ray protection to benefit devil rays?

Many countries still only apply protective measures to manta rays despite all mobulids being highly sensitive to overexploitation. Those countries offering protection for both devil and manta rays include Australia, Brazil, the Member States of the European Union, Israel, Mexico, Ecuador, New Zealand, and the Maldives (*Camhi et al., 2009*; *Whitcraft, O'Malley & Hilton, 2014*; *Council Regulation (EU), 2015*; *CITES, 2015*; *Department of the Environment, 2016*). National asymmetries in protection are still apparent in Peru, the Philippines, United Arab Emirates (UAE), and Indonesia, as these countries afford legal protection to one or both species of manta ray, but not yet to devil rays. We hope that a key outcome of our strategy development is a greater awareness of the need for matching protection and conservation of both devil and manta rays.

International protection for mobulids has expanded relatively rapidly in recent years, and following an initial lag in devil ray protection, the majority of international agreements now protect all mobulids. The first major international action for mobulids came in 2011 with the listing of just the Giant Manta Ray on Appendix I and II of the Convention on the Conservation of Migratory Species of Wild Animals (CMS), obligating the 122 Parties to strictly protect the species and collaborate toward regional conservation. In 2013, the two manta ray species were listed under CITES Appendix II, the world's oldest and largest multilateral environmental agreement with the legal mechanisms in place to hold Parties accountable to trade restrictions. The 183 Parties are thus required to issue permits to export manta rays (or manta ray products) only after demonstrating that they are sourced from legal and sustainable fishing operations (*CITES, 2013*). In 2014, during the 11th Meeting of the CMS Parties, the remaining ten species of mobulid were listed on Appendix I and II (*CMS, 2015*). In 2016, during the 17th Meeting of the Conference of the Parties to CITES, all species of *Mobula* were listed under Appendix II (*CITES, 2016*; Action 6.10). In 2015, the General Fisheries Commission for the Mediterranean (GFCM) became the first Regional Fishery Management Organization (RFMO) to prohibit take of a mobulid species (the Giant Devil Ray). This was followed later that year by a binding measure adopted by the Inter-American Tropical Tuna Commission (IATTC). This IATTC measure aims to prevent targeting, retention, and discard mortality for all mobulid species taken in relevant Eastern Tropical Pacific fisheries, but includes notable exceptions for small-scale operations.

## Can responsible trade and demand reduction help to decrease target fisheries for gill plates?

Trade regulation can lead to both positive and negative outcomes. Trade may cease, or continue at sustainable levels as a result of regulation; or trade may continue illegally, or continue without full regulatory compliance. Trade regulation, therefore, should be promoted and implemented with careful consideration of socio-economic drivers. Sometimes governments find it simpler to ban, rather than regulate, trade (*Vincent et al., 2014*). Regulatory obligations (such a CITES permit processing system) can be challenging to implement, particularly in countries with low capacity for management (*Shepherd & Nijman, 2007*; *Rosen & Smith, 2010*). Counter-intuitive to the intended purpose, complete bans can sometimes stimulate wildlife exploitation by driving it underground and creating circumstances for elevated value. An analysis of mainly terrestrial species policies that changed from allowing a regulated trade at sustainable levels to a complete ban on trade, found that trade volumes increased by 135% in the year prior to the ban (*Rivalan et al., 2007*). The price of rhino horn in Korea rose by 400% two years after a total ban, which fueled a sharp increase in poaching (*Rivalan et al., 2007*; *Biggs et al., 2013*).

One approach to preventing negative conservation outcomes from trade regulation involves attempting to understand relevant socio-economic drivers and associated stakeholder behaviour. Investigations into the mobulid gill plate trade found it to be centered in Guangzhou, China, and involve only a handful of large suppliers (*Whitcraft, O'Malley & Hilton, 2014*; *O'Malley et al., 2016*; Actions 7.3, 7.4, 7.5). Conservation

campaigns aimed at consumer demand reduction in Guangzhou, along with stronger Chinese government policies that inadvertently affect wildlife trade, appear to be reducing demand for gill plates (*O'Malley et al., 2016*, Action 7.7); however, continued monitoring is needed to evaluate and track the success of such consumer behaviour change campaigns. Anecdotal information suggests that trade regulation specific to manta rays may unintentionally increase fishing and trade pressure on devil rays. For example, in Indonesia, where national protection exists for manta rays but not devil rays, fishers have begun to target devil rays to avoid penalties. In this case, fishers claim that devil rays are more challenging to catch than manta rays, and thus were less frequently targeted prior to national manta ray protection (D Adhiasto, pers. comm., 2016).

Another important consideration of regulating mobulid ray exploitation is the effect on livelihoods and food security of fishing communities. Human populations in many tropical coastal communities are growing rapidly, have low income, and rely heavily on fish for protein and income (*Allison et al., 2009*). Mobulids provide a source of income and protein in several developing countries, particularly Indonesia and Sri Lanka (*Fernando & Stevens, 2011*; *Lewis et al., 2015*). Developed nations can increase the effectiveness of conservation measures by helping to facilitate necessary social and economic transitions (*McClanahan et al., 2008*). Under CITES Appendix II, countries can independently assess the sustainability of exports and determine the allowable level of trade, if any. Countries may prohibit landings of mobulids from fisheries based on depletion, high vulnerability, or precaution, regardless of CITES listings. Some mobulid species may be able to support sustainable fisheries, but this has yet to be documented in practice, and would require robust management and enforcement. All fisheries that catch mobulids, including fisheries where catch is incidental, should be monitored, regulated, and minimized when necessary to ensure that these fisheries are sustainable.

## How can the impact of incidental catch on mobulids be reduced?

Incidental catch in both large and small-scale fisheries is a key challenge to the conservation of mobulids, given mortality during capture, evidence for low rates of post-release survival (*Francis & Jones, 2017*), and increased incentives for fisheries to retain these species for the gill plate trade (*Croll et al., 2015*). The few available studies on post-release survival in devil rays show that handling following capture may strongly influence post-release survival (Action 5.8), although more research is needed (Action 5.9). In a study where small (142–238 cm disc width (DW), mean 200 cm DW) Spinetail Devil Rays were tagged without being removed from the water as part of a scientific study, post-release survival was relatively high (*Croll et al., 2012*). In contrast, when large individuals (215–265 cm DW) of the same species were brought on the deck of commercial fishing vessels prior to being tagged and released, post-release survival was low (*Francis & Jones, 2017*). Researchers found that removing these rays from the water caused significant physical strain with potential for post-release mortality. For tuna purse seine fisheries, releasing large rays directly from the brailer (scoopnet that removes the fish from the purse seine), or lifting them out of the brailer using a canvas sling or scoop, is considered best practice; small and medium rays landed on the fishing vessel deck can be carried by their wings to be released (*Poisson et al., 2014*).

The IATTC's 2015 prohibition on the retention, transshipment, storage, landing, and sale of all devil and manta rays in large-scale, Eastern Tropical Pacific tuna fisheries is a significant step toward mobulid protection, but compliance has yet to be evaluated, and exceptions made for small-scale fisheries compromise the conservation goals that drove the original proposal (Action 5.13; *IATTC, 2015*). Effective implementation of associated requirements for reporting mobulid catch data and ensuring safe releases, as well as provisions for technical assistance and capacity building, is critical to improving the outlook for the species in this region (Action 5.3, 5.4, 5.5).

While the IATTC mobulid measure can be seen as progress for some purse seine fisheries, more attention to the incidental catch of mobulids in data-poor trawl fisheries, particularly midwater trawls, is needed. A study on the pelagic megafauna taken incidentally in large-scale European trawl fleets targeting sardinella, sardine, and horsemackerel off the Northwest Africa shelf found that approximately one mobulid is taken for every hour of trawling (*Zeeberg, Corten & de Graaf, 2006*). More research and management is urgently needed to address threats to mobulids from such trawling operations and from many other fisheries (Action 5.6).

## CONCLUSIONS

Research and regulations for both devil and manta rays have recently increased substantially, due in large part to these species' high sensitivity to over-exploitation. Manta rays, however, have been a greater focus for scientific investment and protection, owing to the added factors of perceived charisma and importance to the dive tourism industry. To bridge this charisma gap and encourage equal protection for all mobulids, as warranted by similar life history and threats, we present a global conservation strategy that draws on components from well-established and successful species conservation frameworks. We are hopeful that sustained interest and collaborative implementation initiatives from the full range of stakeholders will improve the outlook for these remarkable fishes.

## ACKNOWLEDGEMENTS

We thank Amie Bräutigam, Martin Clark, Kerstin Forsberg, Sarah L. Fowler, Martin A. Hall, Sarah A. Lewis, and Alessandro Ponzo for their contributions to the workshop. We also thank additional contributors including: Ramón Bonfil, Clinton A.J. Duffy, Rachel T. Graham, Shawn Heinrichs, Rima W. Jabado, Tom Kashiwagi, Andrea Pauly, Glenn Sant, Fabrizio Serena, Melanie Virtue. David W. Sims facilitated our symposium at the Fisheries Society of the British Isles meeting in Plymouth, United Kingdom.

### Funding

Funding provided by Natural Science and Engineering Research Council, Canada (NKD, LNKD), the Canada Research Chairs program (NKD), and the Save Our Seas Foundation project #235 (NKD) and #242 (DF). Support for workshop participants was provided

by the Project AWARE Foundation (AB and Kerstin Forsberg), WildAid (MRH, MPO), and the New England Aquarium (Kerstin Forsberg). The following grant information was disclosed by the authors: Natural Science and Engineering Research Council, Canada; Canada Research Chairs program; Save Our Seas Foundation project: #235, #242; Save Our Seas Save Our Species project: #2013A-058, #2013A-069; US State Department Contribution to IUCN. The funders had no role in study design, data collection and analysis, decision to publish, or preparation of the manuscript.

## Grant Disclosures

The following grant information was disclosed by the authors:
Natural Science and Engineering Research Council, Canada (NKD, LNKD).
Canada Research Chairs program (NKD).
Save Our Seas Foundation: #235 (NKD), #242 (DF).
Project AWARE Foundation (AB and Kerstin Forsberg).
WildAid (MRH, MPO).
New England Aquarium (Kerstin Forsberg).
Save Our Seas Save Our Species project: #2013A-058, #2013A-069.
US State Department.

## Competing Interests

The authors declare there are no competing interests. Opinions expressed herein are of the authors only and do not imply endorsement by any agency or institution associated with the authors.

## Author Contributions

- Julia M. Lawson and Mary P. O'Malley analyzed the data, contributed reagents/materials/analysis tools, wrote the paper, prepared figures and/or tables, reviewed drafts of the paper.
- Sonja V. Fordham and Colin A. Simpfendorfer conceived and designed the experiments, performed the experiments, contributed reagents/materials/analysis tools, wrote the paper, reviewed drafts of the paper.
- Lindsay N.K. Davidson and Daniel Fernando contributed reagents/materials/analysis tools, prepared figures and/or tables, reviewed drafts of the paper.
- Rachel H.L. Walls and Isabel Ender conceived and designed the experiments, performed the experiments, contributed reagents/materials/analysis tools, reviewed drafts of the paper.
- Michelle R. Heupel and Ania Budziak contributed reagents/materials/analysis tools, wrote the paper, reviewed drafts of the paper.
- Guy Stevens and Nicholas K. Dulvy conceived and designed the experiments, performed the experiments, contributed reagents/materials/analysis tools, wrote the paper, prepared figures and/or tables, reviewed drafts of the paper.
- Malcolm P. Francis and Giuseppe Notarbartolo di Sciara contributed reagents/materials/analysis tools, reviewed drafts of the paper.

## Data Availability

The research in this article did not use, analyse, or collect raw data.

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
