# Peer review of "Sympathy for the devil: a conservation strategy for devil and manta rays"

_PeerJ, doi:10.7717/peerj.3027_

## Round 0.1 · original submission · Major Revisions

I have now heard back from three reviewers. All three applauded the amount of work that has gone into this review. At the same time, two offer suggestions on how to improve the current manuscript. In particular, both reviewers 2 and 3 note that this paper includes two different parts - a review/data section and a strategy section. Reviewer 2 requests more explanation of results, while reviewer 3 offers many suggestions and asks questions on many sections of this paper. While PeerJ does not usually accept literature reviews in the traditional sense, in my opinion the topic of your paper is of importance, and your paper does have a data section combined with strategy planning - and consider it to fit within PeerJ's scope. Due to the large amount of comments combined with the large amount of data within your paper, revisions of this scale will take no small amount of time and effort, and hence my decision is 'major revision'.

Reviewer 1 ·

Basic reporting

Check order of references to figures in text and that they refer to the appropriate figure.

Experimental design

It is unclear how the review of the scientific literature on this group is actually implemented in the paper other than a few metrics on increasing attention to studying them and thus more publications and higher citations. I would've appreciated even a general categorization of each paper found by literature review which would thus support the areas the authors later describe as lacking attention (e.g. maturity and reproduction).

Validity of the findings

The maps and tables provide the best summary of the work and table 2, in particular, provides a concise way to communicate the way forward on mobulid research, conservation, and management.

Additional comments

None.

Reviewer 2 ·

Basic reporting

No Comments

Experimental design

Obtaining the information on a difficult group of species was logical through literature search. The quantity of expert knowledge possible at the workshop was limiting but increased through subsequent survey, and, although gaps remained, this was probably to little detriment.

Validity of the findings

Given the information presented and used is largely qualitative, the conclusions are acceptable.

The problem to me is that this is really 2 papers: one on the data collected and the other on the Conservation Strategy. In the Results section, the former is covered between lines 255-271, and the latter is just mentioned between lines 272-275.

Given the distribution and protection data, Figs 2-5 have a huge amount of information which is not described in lines 255-271. One is left to interpret the figures, and I confess, they are very confusing: is the issue of too much information on one figure? (There is also a personal issue that I am slightly colour blind, so discrimination between tones is simply too challenging).

The Conservation Strategy is excellent, prompting only 2 points:

1/ There are mentions within it of Actions in specific countries, which do not relate back to any points made in the text: does this matter?
2/ The Conservation Strategy is going to be of such importance for conservation purposes, that I question whether it is not going / should be a standalone product

Additional comments

This is a tremendous effort and much needed initiative, but I think more explicit interpretation of the accumulated information would be beneficial. I suspect some of the information in the Figures could be handled through tables - less colourful but easier to use.

Annotated reviews are not available for download in order to protect the identity of reviewers who chose to remain anonymous.

·

Basic reporting

The introduction to this paper is tenuous, with its emphasis on traditional Chinese medicine. In fact, codified TCM & TCM practitioners don't really used these products. The introduction might have been more fittingly directed at conservation strategy. See my general comments to authors.

Experimental design

This paper has little primary research. It is rather an evaluation of existing knowledge and the development of a conservation strategy. It largely doesn't conform to the criteria outlined for this section, in terms of a defined research question, good technical investigation etc. See my general comments to the authors.

Validity of the findings

The conclusions are in the form of one figure and two tables. The second table is a conservation wish list, albeit a thoughtful wish list. I found little connection between the putative methods/results and the discussion, which covered interesting material in a manner quite detached from the paper thus far. See my general comments to the authors.

Additional comments

This paper sets out to explore the current status and options for devil and manta rays, two taxa of notable conservation interest and importance.

The paper reads very much like a workshop report / conservation essay rather than a primary contribution. There are four sections of methods and results: literature tally, list of expertise, mapping ranges against protection/threats and a conservation strategy. The first two of these are of negligible interest or value outside a workshop report. The third is useful but needs to be consolidated into one figure (Fig 2) and one table (Figs 3-5). The conservation strategy is rather more a wish list than a plan, and should be revised accordingly.

I am aware that mobulid rays may well be proposed for CITES Appendix II listing at the next CITES CoP, making any output on these fishes timely. To that end, I have offered detailed input on this manuscript as I hugely value the conservation effort invested by this team for these taxa.

Abstract
- This needs a rewrite to convey information not process
- Methods and results don’t match here
o you said that you “reviewed the state of the development of scientific knowledge and capacity for these species, and summarise the geographic ranges, fisheries and national and international protections for these species.” – so you should provide some of this information in the results – you have space
o you said you were going to “develop the Global Devil and Manta Ray Conservation Strategy, specifying a vision, goals, objectives, and actions to advance the conservation of both devil and manta rays.” – so tell us something
- abstract says very little – end of it is all about “we examined” but should tell us what you found / deduced
- “similarities in sensitivity and appearance” – to what ?

Introduction
- This paper should revolve around conservation planning as a theme and not around an exaggerated link to TCM
- first line needs a reference – “has elevated the demand for Traditional Chinese Medicine” – I don’t have one – seems to be an inference and not a certainty
- some factors are also driving reduced TCM use – urbanization, westernization, TCM training that does not include animal products – this also needs to be considered
- what part of sharks is used in TCM ? only cartilage I think ? where are the seahorses here ?!
- line 89 should be the topic sentence for the next paragraph on gill rakers
- your first paragraph and opening focus on TCM is misleading – as you outline, the use of gill plates is not traditional Chinese medicine – it would be better labeled as Chinese folk medicine – and the first paragraph should be on the growth of the Chinese population and incomes, not on tenuous changes in TCM
- if the rays are commonly a targeted secondary catch, then the emphasis on bycatch needs to be reconsidered
- if you are going to talk targeted then bycatch, keep that order throughout the paragraph
- “Nine species of devil ray (genus Mobula) and two species of manta ray (genus Manta) make up the subfamily Mobulinae”. I would put this parag above the two that precede it. But the topic sentence should be the one at the end of the parag “The life history and ecological traits of mobulids make them highly sensitive to overexploitation.”
- “Protections for mobulids have expanded relatively rapidly over the past decade” – very long parag and should be in your results
- “Conservation planning” – this paragraph didn’t go anywhere – you should introduce this tool in its own parag and link it to other taxa and experiences

Methods
- how did paper gathering and the workshop connect ?
- the first parag is too long and overly detailed – cut by half at least
- “The IUCN SSG convened the Global Devil and Manta Ray Conservation Strategy Workshop in Durban, South Africa from 9-12 June 2014, and initiated a survey among a wider network of devil and manta ray experts” – this parag should be in the intro as it is really what this paper is about
- line 212 – this parag can be shortened – e.g. lose this sentence “The Global Sawfish Conservation Strategy argued …” – it has no role here
- line 224 “where possible” – how was that determined ? – when wouldn’t it be possible ?

Results
- this analysis of the number of papers seems meaningless here – delete the section or set it up much better in the intro & methods to anticipate why this matters
- “The current state of devil and manta ray expertise” – this parag belongs in the methods – attendance ≠ state of expertise
- line 255 subtitle is ranges, fisheries and protection – but you then go almost at once to protection
- line 257- isn’t this Fig 2 ?
- fig 2 – why do j & k have different colours to the other maps ?
- fig 3 – expertise – this is not a result – it’s merely an input to the workshop or a method – and it would be enough to say that x MFAs were represented at the workshop
- figs 4 & 5 doesn’t work – too busy – very unclear what you are trying to show – change to table with country/archipelago/sea (as seems best) as rows against no. manta spp., no. devil spp, CITES yes/no, CMS yes/no, targeted fisheries yes/no, bycatch yes/no , fishing restriction yes/no, trade restriction yes/no & whatever else
- fig 5 – why don’t all CITES Parties show a trade restriction ? – why do you have circles around Hawaii and Florida which are hardly small island nations ? – in fact are any circles around small island nations ?
- Table 1 is useful but why did you go with reverse chronological order rather than alphabetic or chronological order ?
- Lines 258-262 – these sentences are repetitive
- I worry that this “strategy” is a wish list rather than a plan. I won’t presume to evaluate the specific actions, devised by the top experts on rays. The strategy itself comes across, however, a huge list of actions, with no apparent prioritization, metrics, timelines or actors. Yet priorities must be set. I suggest that the team add a problem evaluation. There is a need to sort actions by urgency, importance, feasibility, probability, cost etc and create a plan accordingly. What are the sweet spots where a limited amount of investment could yield big results ? Then you are getting closer to strategy. And, yes, I know this is modeled on the sawfish strategy.

Discussion
- this discussion does not follow from the results – rather it seems to be an opinion piece – no connection with all that went before – and rather too many disparate bits – e.g. line 303
- policy has been put in place but I doubt you can comment on whether protection has improved yet
- line 295 “conservation successes of manta rays” - what are these ? – where are the data ? – activities & policies, yes, but success ? measured how ?
- line 361 – why this parag – is there a proposal to uplist ?
- line 374 – such claims of success are common but generally unsubstantiated – and this is unpublished data so should not be cited here – so this parag is poorly supported and should be reduced to one sentence about trying to reduce demand
- line 387 - these low-volume high-value fisheries “may support livelihoods and contribute to food security”- really need some data on this before going much farther with it

General comment
- The structure of the paper is hard to follow with some real jumbling between introduction, methods and results
- be consistent in paper – mantas then devils or vv – in all sections
- language needs a bit of tightening and grammar check – some small typos throughout (e.g. line 188 “gage”)

---

## Round 0.2 · Minor Revisions

I have heard from one reviewer, who was very positive regarding this revised version. I myself have also gone over this new version, and find it much easier to read and understood; you should all be commended for your hard work. I have a few small changes and one suggestion, detailed in the attached PDF. Thus, my decision is "minor revisions" with the emphasis on "minor". I look forward to seeing a revised version.

Reviewer 2 ·

Basic reporting

The paper has undergone radical restructuring based on the comments of all 3 Reviewers of the original submission.

I have tested the current text against my comments on fundamental issues, and the basic problem (to me) that it was 2 papers, one on collected and collated inforamtion and the other on a conservation strategy is now resolved.

Experimental design

Given the situation re the species, the experimental design is perfectly satisfactory, as i noted before.

Validity of the findings

Very sound, especially for subject on which this paper is really the first systematic foray into the species' status and conservation.

---

## Round 0.3 · accepted · Accept

The manuscript has been well revised, and is now ready to be published. I look forward to seeing the published version!